# Parkinson’s Disease Subtyping Using Clinical Features and Biomarkers: Literature Review and Preliminary Study of Subtype Clustering

**DOI:** 10.3390/diagnostics12010112

**Published:** 2022-01-04

**Authors:** Seung Hyun Lee, Sang-Min Park, Sang Seok Yeo, Ojin Kwon, Mi-Kyung Lee, Horyong Yoo, Eun Kyoung Ahn, Jae Young Jang, Jung-Hee Jang

**Affiliations:** 1Global Health Technology Research Center, College of Health Science, Korea University, Seoul 02841, Korea; aksska82@korea.ac.kr; 2KM Data Division, Korea Institute of Oriental Medicine, Daejeon 34054, Korea; kiro123@kiom.re.kr (S.-M.P.); ekahn@kiom.re.kr (E.K.A.); 3Department of Physical Therapy, College of Health Sciences, Dankook University, Cheonan 31116, Korea; yeopt@dankook.ac.kr; 4KM Science Research Division, Korea Institute of Oriental Medicine, Daejeon 34054, Korea; cheda1334@kiom.re.kr; 5Biological Resource Center (BRC), Korea Research Institute of Bioscience and Biotechnology (KRIBB), Jeongeup-si 56212, Korea; miklee1010@kribb.re.kr; 6Clinical Trial Center, Daejeon Korean Medicine Hospital, Daejeon 35235, Korea; horyong.yoo@gmail.com; 7School of Electrical, Electronics and Communication Engineering, Korea University of Technology and Education (KOREATECH), Cheonan 31253, Korea

**Keywords:** biomarker, clinical subtyping, cluster analysis, neurodegenerative disorders, Parkinson’s disease

## Abstract

The second most common progressive neurodegenerative disorder, Parkinson’s disease (PD), is characterized by a broad spectrum of symptoms that are associated with its progression. Several studies have attempted to classify PD according to its clinical manifestations and establish objective biomarkers for early diagnosis and for predicting the prognosis of the disease. Recent comprehensive research on the classification of PD using clinical phenotypes has included factors such as dominance, severity, and prognosis of motor and non-motor symptoms and biomarkers. Additionally, neuroimaging studies have attempted to reveal the pathological substrate for motor symptoms. Genetic and transcriptomic studies have contributed to our understanding of the underlying molecular pathogenic mechanisms and provided a basis for classifying PD. Moreover, an understanding of the heterogeneity of clinical manifestations in PD is required for a personalized medicine approach. Herein, we discuss the possible subtypes of PD based on clinical features, neuroimaging, and biomarkers for developing personalized medicine for PD. In addition, we conduct a preliminary clustering using gait features for subtyping PD. We believe that subtyping may facilitate the development of therapeutic strategies for PD.

## 1. Introduction

Parkinson’s disease (PD), a common chronic progressive neurodegenerative disorder, is characterized by α-synuclein aggregations and neuronal loss in the substantia nigra, which results in striatal dopamine deficiency [1]. The pathophysiology of PD involves multiple neurotransmitter deficiencies resulting in multisystem neurodegeneration, contributing to a clinical phenotyping variability [2,3]. PD is associated with a broad spectrum of clinical symptoms, including motor and non-motor symptoms [1,3]. These multiple clinical symptoms are associated with the progression of PD. In other words, the progression of PD is driven by the combination of increasing severity of non-motor and motor symptoms, complications, and poor response to standard therapy [1]. The clinical manifestations, course of progression, and biomarker profiles in PD vary widely from person to person [4]. Due to the clinical heterogeneity of PD, a personalized approach with a holistic perspective is needed for each patient [3]. For personalized medicine, subtyping according to the common clinical characterization is required. Therefore, a classification for subtyping PD based on clinical, pathological, genetic, and molecular features and biomarkers for distinguishing each subtype should be developed [5]. Here, we have focused on the clinical manifestations, including gait disturbance, neuroimaging, and molecular markers for subtyping PD.

Among the motor symptoms of PD, the gait disturbance increases as the disease progresses [6,7]. The gait disturbance in PD is characterized by decreased steps per minute, stride length, and velocity, and increased double support time and gait variability [8,9]. In a study using kinematic patterns for gait analysis in PD, a difference in kinematic parameters during gait has been reported in patients with PD and healthy elderly subjects [10]. As such, by assessing gait patterns, patients with PD can be distinguished from healthy elderly subjects. Moreover, there are differences in gait even among patients with PD. However, subtype classification studies using gait patterns in PD are still limited.

In research settings, neuroimaging has been used in the differential diagnosis of PD for decades. For a long time, neuroimaging research in PD has focused on the dopaminergic system; however, in recent years, there has been an increase in techniques that are based on magnetic resonance imaging (MRI) and functional imaging (positron emission tomography (PET) and single-photon emission computed tomography (SPECT)) [11,12]. SPECT and PET are effective in distinguishing between degenerative and non-degenerative causes of PD [13]. MRI also provides information about the degeneration in PD. MRI and PET can differentiate between PD and atypical Parkinsonism, but they may require advanced tools for enhancement [14]. Dopaminergic and serotonergic PET can be used to monitor the progression of PD and its motor and non-motor symptoms and complications; however, few of these findings have been applied to clinical practice. Hybrid PET-MRI technology has dramatically altered PD imaging, but image reconstruction must be addressed before its use in research and clinical settings. The high cost prevents the transfer of neuroimaging from research to clinical practice.

Various studies have been attempted to develop reliable diagnostic and prognostic biomarkers for PD through genetic, biochemical, and transcriptomic analyses. Linkage and next-generation sequencing studies have revealed the genetic landscape of PD [15]. Investigations on the pathobiology of identified genetic risk factors that may result in a difference in onset, symptoms, and progression of PD are still ongoing. Biochemical markers in the blood or cerebrospinal fluid (CSF) are expected to facilitate an early diagnosis and assess the severity of PD [16]. The concentration levels of these markers are associated with clinical symptoms and neuroimaging features. Transcriptome analysis based on high-throughput sequencing technology allows the characterization of genome-wide expression levels in patients with PD [17]. Transcriptomic signatures in patients with PD have been reported to be distinct from those observed in healthy controls (HCs); similar differences have been observed among subgroups of patients with PD.

Meanwhile, previous studies using a single feature for PD subtyping have limited applicability in clinical practice. Additionally, the biomarker profiles in PD differ among individuals [4], and there is a growing number of failed attempts at establishing a simplistic, single-target approach towards the treatment of PD. Therefore, the symptomatic and pathological differences between patients with PD should be addressed [5]. Subtyping according to clinical manifestations and objective biomarkers is necessary for an accurate diagnosis of PD occurrence and progression. We investigated the classification of PD subtypes using clinical symptoms, neuroimaging, and molecular markers in previous studies. Additionally, we conducted a pilot study on whether preliminary clustering using gait features is possible for classifying PD subtypes. As PD is a disease associated with complicated motor and non-motor symptoms, the subtyping should integrate features, such as gait disturbance, neuroimaging, and molecular markers, among others. Therefore, here, we aimed to develop an integrated classification model for PD subtypes (Figure 1) based on the results of the pilot study.

## 2. Classification of PD

The age of onset, clinical phenotypes, and disease severity or neuropathological alterations have been reported as the criteria for classifying PD subtypes [3]. In this section, we review PD subtypes using clinical features, neuroimaging, and molecular markers.

### 2.1. Classification of Subtype According to Clinical Features

The clinical symptoms of patients with PD are complex, encompassing a broad spectrum of symptoms, including cardinal motor symptoms (tremor, rigidity, bradykinesia, postural instability, and gait disorders) and non-motor symptoms (sleep–wake cycle disorders, cognitive impairment, mood and affective disorders, autonomic dysfunction, and sensory symptoms and pain) [1]. The variability in the clinical phenotype of PD is suggested to represent various subtypes of the disease [18], implicating the difference in their pathogenetic hypotheses and therapeutic strategies [19]. Previous studies classifying PD subtypes using clinical phenotypes have suggested the following four subtypes: type I, tremor-dominant (TD); type II, postural instability and gait difficulty (PIGD); type III, rigidity-dominant; type IV, axial-dominant; type V, appendicular-dominant. This classification was based on the presence of motor symptoms as the dominant criteria [20]. Recently, it has been observed that the non-motor symptoms are playing an increasingly important role in the clinical heterogeneity of PD [21], and subtype classification studies utilizing the dominant clinical symptom and disease severity as criteria, considering both motor and non-motor symptoms, have been reported [18,19,21]. In three previous studies, based on the severity of symptoms, the disease was classified as mild [21] or mild motor and non-motor disease [18], and severe [21] or severe motor and non-motor disease with poor psychological well-being [18]; based on the dominance of symptoms, it was classified as non-motor-dominant [19,21] or poor psychological well-being, rapid eye movement sleep behavior disorder (RBD), and sleep [18], motor-dominant [19,21], poor posture and cognition [18], and severe tremor [18], benign pure motor [19], and benign mixed motor–non-motor [19]. Several clinical studies have attempted to classify the clinical subtypes of PD from cluster analysis using a longitudinal database including cohorts (Table 1).

#### 2.1.1. Subtype Classification Using Motor Symptoms and Marker Investigation

Several studies have investigated markers for distinguishing subtypes in PD, classified by the clinical phenotype (Table 1). Previously, PD was often divided into two subtypes based on the dominant motor symptom: TD and PIGD [22]. According to the subtype, the disease progression was closely related to clinical symptoms [23,24]. Therefore, many studies were conducted to develop markers and compare the characteristics of the TD and PIGD subtypes. On comparing the severity of clinical symptoms between the two distinct subtypes, PIGD was associated with more severe symptoms and a higher mortality rate than TD [25,26,27,28,29,30]. In addition, several studies have attempted to classify PD subtypes based on gait characteristics [23,24]. The gait and balance disturbances, such as reduced gait speed, shorter strides, increased stride variability, and increased stride irregularity, and performance-based tests for mobility, balance, and fall risk using the objectively quantified assessment, were more severe in the PIGD subtype [23,31]. Wu et al. compared spatiotemporal and kinematic parameters of gait by dividing 86 patients with PD into PIGD (*n* = 56) and TD (*n* = 30) groups [32]. In terms of spatiotemporal parameters, the PIGD group showed shorter stride length, increased stride time, and higher stride length variability when compared to the TD group. In terms of the kinematic parameters, the ankle joint angle and toe-off angle during gait were significantly decreased in the PIGD group when compared to the TD group, and there was no significant difference between the two groups in the knee joint, hip joint, and heel strike angle. As a result, the PIGD group was more severely affected by spatiotemporal parameters during gait; in particular, the motor deficit was more severe in the distal joint. As a behavioral marker for subtyping PD, the different aspects of interoceptive, somatosensory sensations related to non-motor phenomena, and processing deficits in PIGD and TD subtypes, were demonstrated. The interoceptive accuracy and sensibility were more reduced in the patients with the TD subtype than in those with the PIGD subtype [33].

Freezing of gait (FOG) is a common symptom in patients with PD; it is defined as restriction in the forward locomotion despite efforts to move forward [34,35]. Factor et al. classified patients with FOG into responsive FOG (RFOG) and unresponsive FOG (URFOG) based on the patients’ response to levodopa, and compared them with PD patients without FOG [35]. The URFOG group had a higher age of disease onset and a higher unified Parkinson’s disease rating scale score than the RFOG group [35]. For cognitive function, the patients with URFOG and RFOG had a lower score than those without FOG. In terms of visuospatial performance (visuospatial domain and executive functioning domain), the URFOG group showed lower performance than the RFOG and no FOG groups [35]. Conversely, the frequency of general hallucinations was higher in the RFOG group (53%) compared to the URFOG (25%) and no FOG (20%) groups. In terms of dyskinesia, both the URFOG (44%) and RFOG groups (70%) had more frequent dyskinesia compared to the no FOG group (24%) [35]. This indicates that, depending on the response to levodopa in PD patients with FOG, there may be differences in the patient’s cognitive, motor, and perceptual functions; moreover, PD patients can be sub-grouped using various other methods.

To develop markers, the differences in the pathophysiological mechanisms, dysfunction in specific brain regions [36,37], and serum antioxidative property [38] in PD were investigated. To investigate the brain connectivity dysfunction in the TD and PIGD subtypes, the difference in the PD-related degeneration of brain hubs in the TD and PIGD subtypes was analyzed using 3T resting-state functional MRI [36]. The topological organization of brain functional networks was altered [39], and comprehensive disruption in brain regions, including the basal ganglia, cerebellum, superior temporal gyrus, pre- and postcentral gyri, inferior frontal gyrus, middle temporal gyrus, lingual gyrus, insula, and parahippocampal gyrus, was found in both subtypes of PD. Notably, the PIGD subtype displayed more disrupted hubs in the cerebellum. The cerebellum demonstrated dopaminergic degeneration, α-synuclein deposition, and aberrant projections from the basal ganglia in PD. Therefore, more severe cerebellum disruptions in the PIGD subtype can explain the greater loss of functional connectivity [36]. In another study, both subtypes of PD had shown a change in the regional homogeneity values in the basal ganglia–thalamus–cerebral cortex circuit and extensive abnormalities in the basal ganglia, thalamus, limbic system, parietal lobe, occipital lobe, and frontal lobe. In particular, regional homogeneity values in the parahippocampal gyrus were more increased in the TD subtype than in the PIGD subtype, thereby indicating a compensatory slow progressive cognitive decline in the TD subtype [37]. To investigate the correlation between the motor subtype and serum antioxidative property, the differences in serum bilirubin (an important natural antioxidant and a major contributor to the total antioxidant capacity of plasma [40]) concentrations were evaluated in patients with the motor PD subtype and HCs. Total bilirubin and indirect bilirubin concentrations were significantly lower in PD patients than in HCs and in the PIGD subtype than in the TD subtype. The decrease in indirect bilirubin concentrations may result in the lack of an endogenous defense system required to prevent oxidative stress from the damage and destruction of dopaminergic cells in the substantia nigra of the motor subtype [38].

In addition to the PIGD and TD subtypes, four motor progression phenotypes were identified using the Hoehn and Yahr scale at the baseline, 12 months, and 36 months milestones: secondarily progressive PD, early progressive PD, non-progressive PD, and minimally improving PD. SFT, serum insulin-like growth factor-1, CSF α-synuclein, and dopamine transporter (DaT)-SPECT-derived basal ganglia striatal binding ratios were suggested as possible motor progression biomarkers [41].

#### 2.1.2. Subtype Classification Using Non-Motor Symptoms and Marker Investigation

The motor symptoms are considered a core feature in PD. However, with emerging non-motor symptoms, its diagnostic criteria were revised to include non-motor symptoms in the core parameters [21]. For the diagnosis, non-motor symptoms in PD were classified into the following subtypes: amnestic and non-amnestic mild cognitive impairment (MCI). The cortical thickness, hippocampal volume, white matter integrity, and striatal dopamine nerve terminal integrity between the two subtypes were not significantly different. However, cognitive impairment, dementia conversion, and functional connectivity in the left parietal cortex with the salience network were increased in PD patients with amnestic MCI [42]. PD-amnestic MCI may exhibit different functional correlations of the substantia nigra without concomitant structural abnormalities, and this difference may affect the cognitive prognosis and PD dementia conversion risk.

#### 2.1.3. Subtype Classification Using Motor and Non-Motor Symptoms and Marker Investigation

According to a previous study, the most critical determinants for classifying subtypes and predicting the prognosis of PD were non-motor symptoms, including cognitive status, RBD, and orthostatic hypotension [43]. Thus, the subtyping in PD should not be limited to motor symptoms; several studies have subtyped via clustering motor and non-motor features [21].

To investigate the association of dopaminergic dysfunction in the putamen, caudate, and striatum with the clinical phenotype (motor and non-motor), a non-hierarchical cluster analysis including motor and non-motor data of PD patients was performed using ^123^(I)-FP-CIT SPECT (iodine I 123–radiolabeled 2β-carbomethoxy-3β-(4-iodophenyl)-N-(3-fluoropropyl) nortropane) SPECT images [44]. Three different subtypes were identified: subtype 1 showed the lowest motor and non-motor burdens, possibly indicating its benign nature; groups 2 and 3 displayed similar motor disability but differed from each other in the presence of additional non-motor features, including apathy and hallucinations. ^123^(I)-FP-CIT binding values paralleled motor disability burden; the non-dopaminergic system was possibly associated with the non-motor variability in PD [44].

In addition to the basal ganglia and midbrain, the cerebellar changes in different PD subtypes were compared using MRI [45]. The cerebellar gray matter atrophy was more commonly found in PD subtypes showing depression and anxiety than in the motor-dominant subtypes, thus suggesting a possible role of the cerebellum in the depressive and anxious symptoms in PD [45].

To estimate the course, prognosis, and survival of PD, clinical subtypes considering the severity and rate of longitudinal progression of neuropathologies have been reported. The motor and non-motor symptoms were comprehensively assessed at the baseline and reassessed at the follow-up. In three studies, three distinct subtypes were classified: mild motor-predominant [4,46] or slow progression [43], diffuse malignant, and intermediate. The subtype clustering was based on non-motor features (MCI, RBD, dysautonomia, depression, and anxiety) and motor symptom scores [43]. The motor/slow progression subtype was characterized by either predominant motor manifestations [43] or motor and all three non-motor scores below the 75th percentile [4,46]; it was associated with a favorable course of disease [43]. The diffuse/malignant subtype was characterized by more severe motor symptoms and prominent non-psychiatric disorders [43] or with motor score plus ≥ 1/3 non-motor score above the 75th percentile or all three non-motor scores above the 75th percentile [4,46]; the subtype was associated with rapid malignant progression [4,43] and reduced survival [46]. The intermediate subtype was characterized by motor features similar to the motor/slow progression subtype and moderate/intermediate non-motor symptoms [4,43,46]; it was associated with disease progression higher than the motor/slow progression subtype [43]. The diffuse malignant subtype was more likely to have MCI, orthostatic hypotension, RBD [43], prominent dopaminergic deficit, atrophy in PD-specific brain networks, and an Alzheimer’s disease (AD)-like CSF profile [4]. Although the diffuse malignant subtype progressed rapidly and showed an AD-like CSF profile, the neuropathological findings, such as staging of Lewy body dementia and AD-related pathology at post-mortem, did not differ between subtypes [46]. In a longitudinal cohort study utilizing an automated deep learning algorithm, three subtypes were identified using motor and non-motor assessments, biospecimen examinations, and neuroimaging markers [47]. Subtype I was characterized by a moderate functional decay in motor ability, stable cognitive ability, and significantly lower CSF t-tau levels. Subtype II was characterized by a mild functional decay in both motor and non-motor symptoms. Subtype III was characterized by a rapid progression of both motor and non-motor symptoms; it had the lowest DaTScan striatal binding ratio value in the caudate and putamen, indicating a more severe disease course [47]. Blood biomarkers, including apolipoprotein A1 (lower ApoA1 correlated with DAT deficit), C-reactive protein (CRP; markers of a proinflammatory state), uric acid (role as an antioxidant and free-radical scavenger), and vitamin D (neuroprotective effect in animal models), predicted prognosis for motor and non-motor symptoms in PD subtypes derived from the clinical features [48]. The patients with the severe motor disease subtype had poor psychological well-being and sleep, reduced apolipoprotein A1 levels, and raised CRP levels. The proinflammatory biomarker profile (reduced apolipoprotein A1 and raised CRP) was significantly associated with the severe motor and non-motor disease phenotype [48]. Although the role of uric acid levels as a biomarker in identifying different subtypes of PD was not well explained in the above-mentioned study [48], another study suggested that higher serum uric acid levels were associated with the tremor motor subtype and less fatigue in early PD and could be utilized as an important biomarker for specific motor features [49].

Taken together, pathological results, biospecimen examinations, neuroimaging, and prognosis of the disease tended to be poor in subtypes with severe motor and non-motor symptoms.

**Table 1 diagnostics-12-00112-t001:** The classification of subtypes according to the clinical features in patients with Parkinson’s disease (summary).

Criteria of Classifier	Classification of Subtype	Other Variables of Classifier Profile	Findings	Reference
Motor symptoms	TD PIGD	Non-motor and motor symptoms; mortality	More severe non-motor symptoms (cognitive impairment, hallucinations, psychosis, sleep impairment, fatigue, urinary disturbance) [25,26,27] and QOL ↓ [27] in PIGD More severe motor symptoms (H&Y, UPDRS-motor) [29] in PIGD Mortality ↑ [28] in PIGD Severe olfactory impairment in TD [30]	[25,26,27,28,29,30]
	TD PIGD	Gait pattern using a single body-fixed sensor under single and dual task; balance; fall risk [23] Spatiotemporal parameter under two conditions (unobstructed walking and obstacle avoidance) [31]	Gait speed ↓, stride L ↓, stride variability ↑, stride regularity ↓, performance test score ↓ in PIGD [23] Stride L ↓, velocity ↓, double support ↑ in PIGD and stride velocity ↓ in PIGD and TD during unobstructed walking [31] Trailing toe clearance ↓, leading and trailing velocity ↓, leading crossing step width ↑ in PIGD during obstacle avoidance [31]	[23,31]
	TD PIGD	IMU sensor; spatiotemporal parameter, kinematic parameter	Stride length ↓, stride time ↑, step length variability ↑ Cadence ↑, ankle joint ROM ↓, toe-off angle ↓ in PIGD	[32]
	TD PIGD	Behavioral marker	Interoceptive accuracy and sensibility↓ using heat beat perception task in TD	[33]
	RFOG URFOG No FOG	UPDRS, Mini-mental status exam, Visual hallucinations; Scale for the Assessment of Positive Symptoms, Comprehensive battery of neuropsychological measures	UPDRS ↑ in URFOG compared with RFOG, MMSE ↓ in URFOG and RFOG compared with no FOG, Visuospatial performance ↓ URFOG compared with RFOG and no FOG, Dyskinesia ↑ in URFOG and RFOG compared with no FOG	[35]
	TD PIGD	MRI [37], fMRI [36]	More distrusted hub in cerebellum in PIGD [36] ReHo value ↑ in right para-hippocampal gyrus in TD [37] * Compensatory performance slow progressive cognitive decline	[36,37]
	TD PIGD Intermediate	Total bilirubin, IBIL, Direct bilirubin in serum	IBIL ↓ in PD than control, IBIL ↓ in PIGD than TD * antioxidative property of IBIL	[38]
	SPPD EPPD NPPD MIPD	Serum, CSF, neuroimaging	Differentiated NPPD from EPPD: Serum IGF1, SFT HVLT-R Delayed Recall, HVLT-R Retention, Mean Striatum SBR, Mean Caudate SBR, and Mean Putamen SBR Differentiated NPPD from SPPD: Serum IGF1Differentiated NPPD from MIPD: CSF αSyn, Benton Judgement of Line Orientation Test	[41]
Non-motor symptoms	PD-aMCI (amnestic MCI) PD-naMCI (non-amnestic MCI)		Dementia conversion risk ↑, cognitive decline in frontal/executive function ↑, functional connectivity in the left posterior parietal region ↑, memory domain score ↓ in PD-aMCI	[42]
Motor and non-motor symptoms	Lowest motor and non-motor Motor disability Motor disability with apathy and hallucination	Dopaminergic dysfunction measured by ^123^(I)-FP-CIT SPECT scan	Motor disability burden paralleled with dopaminergic dysfunction and negatively correlated with depression	[44]
	Akinetic/rigidity-predominant tremor-predominant non-motor (dPD, aPD, coPD, nPD)	MRI	GMV ↓ in the left Crus I in dPD GMV ↓ in the tonsil and the right lobule VIII in aPD than nPD GM atrophy including the tonsil, the left lobule VIII, the right lobule VI, the left Crus I, vermis IV, and V in coPD than HC	[45]
	PIGD tremor Mixed non-motor	Serum uric acid	Serum uric acid ↑ in tremor subtype Serum uric acid ↓ in mixed motor subtype	[49]
	Mild motor predominant [4,46] or slow progression [43] Diffuse malignant Intermediate	Lewy pathology and AD-related pathology [46]; CSF amyloid-β and atrophy using MRI [4]	Disease milestones development risk ↑ and survival ↓ [46]; level of CSF amyloid-β and amyloid-β/total-tau ratio↓ and whole brain atrophy ↑ [4]; MCI ↑, orthostatic hypotension ↑, RBD ↑ and rapid progression [43] in diffuse malignant subtype	[4,43,46]
	Subtype I (Mild baseline, moderate motor progression) Subtype II (Moderate baseline, mild progression) Subtype III (Severe baseline, rapid progression)	Clinical information (motor and non-motor assessment), biospecimen examinations, neuroimaging using a deep learning algorithm, LSTM	CSF t-tau level ↓ in subtype I Subtype II DaTScan SBR value ↓ in subtype III	[47]
	Fast motor progression Mild motor diseaseSevere motor diseaseSlow motor progression	Apolipoprotein A1, CRP, uric acid, vitamin D [48]	Apolipoprotein A1 ↓, CRP ↑ in severe motor disease, poor psychological well-being, and poor sleep with intermediate motor progression [48]	[48,50]

TD, tremor-dominant; PIGD, postural instability and gait difficulty; UPDRS, unified Parkinson’s disease rating scale; QOL, quality of life; IMU, inertial measurement unit; ROM, range of motion; RFOG, responsive freezing of gait; URGOG, unresponsive freezing of gait; MMSE, mini mental status examination; MRI, magnetic resonance imaging; fMRI, functional magnetic resonance imaging; IBIL, indirect bilirubin; dPD, depressive but not anxious; aPD, anxious but not depressive; coPD, comorbid depressive and anxious (*n* = 8); nPD, without depressive or anxious symptoms; GMV, gray matter volume; SPPD, Secondarily Progressive PD, H&Y progression between V04 and V08; EPPD, Early Progressive PD, H&Y progression between V0 and V04; NPPD, Non-Progressive PD, no H&Y progression; MIPD, Minimally Improving PD; DaTScan SBR, Striatal Binding Ratio, ReHo, regional homogeneity; IGF1, insulin-like growth factor 1; SFT, serum insulin-like growth factor-1; HVLT-R, Hopkins verbal learning test—revised; MCI, mild cognitive impairment; aMCI, amnestic MCI; naMCI, non-amnestic MCI; 123(I)-FP-CIT SPECT, iodine I 123–radiolabeled 2β-carbomethoxy-3β-(4-iodophenyl)-N-(3-fluoropropyl) nortropane SPECT; RBD, rapid eye movement sleep behavior disorder; CRP, C-reactive protein.

### 2.2. Neuroimaging Biomarkers

Several studies have attempted to establish biomarkers for an early diagnosis and to monitor PD progression (Table 2) [51]. In the context of neuroimaging, PET [52,53] and SPECT [54] can accurately detect PD. However, these methodologies are based on detecting the loss of dopaminergic neurons, which may make them increasingly less sensitive to disease progression. It is desirable to diagnose the disease early, before this regression occurs and motor symptoms begin.

MRI-based techniques, such as structural and morphometric MRI, diffusion-weighted imaging, and magnetic resonance spectroscopy, are representative neuroimaging methods for the differential diagnosis and classification of PD. Structural MRI can also reveal structural changes in the brain, such as cortical characteristics and volume reduction. Through diffusion tensor imaging, structural differences in various brain regions can be observed and potential early PD indicators can be identified.

For example, structural MRI showed that abnormal T2 hypointensities in multiple-system atrophy of parkinsonian-type patients could distinguish these individuals from PD patients with a sensitivity of 88% and specificity of 89% [63]. The PIGD subtype is mainly associated with white matter lesions [64]. Diffusion differences in voxel-based anisotropy, axial diffusion, and radial diffusion were compared in PD patients with PIGD and non-PIGD subtypes [56]. A more significant loss of white matter integrity was found in PIGD subtypes than in non-PIGD subtypes [56].

A functional MRI (fMRI) study evaluated functional brain network alterations in the striatum subregions of early PD patients and HCs [65]. Compared with the control group, the PD group showed reduced functional connectivity in the mesangial–striatal and cortical pleural loops. These results demonstrate the reduced functional integration of neural networks, including the striatum, mesolimbic cortex, and sensorimotor regions, thus suggesting that general disconnection of the mesolimbic–striatal loop is associated with early clinical non-motor functions in PD.

In another study, based on 3T resting-state functional MRI, the intrinsic functional connectivity patterns of forebrain networks were investigated according to PD subtypes [36]. In both the TD and PIGD subtypes, inclusive interruptions were found, mainly in the basal ganglia, cerebellum, superior temporal gyrus, anteroventral gyrus, inferior frontal gyrus, middle temporal gyrus, lingual gyrus, islet, and hippocampus [36]. Additionally, the PIGD subgroup had more hub disruption in the cerebellum than the TD subgroup. These results suggest that the pathophysiological mechanisms of neuronal dysfunction differ between PD subgroups.

In addition, neuroimaging has revealed the pathological substrate of motor symptoms in PD. Morphometric MRI showed that FOG was associated with cortical gray matter loss [66] and decreased fMRI blood oxygenation level-dependent signal in the striatal and extrastriatal regions during virtual reality gait tasks [67]. Glucose metabolism PET imaging in PD patients showed metabolic degradation in the striatum and parietal cortex, which was associated with gait arrest [68].

Studies to date have shown that gait impairment is associated with disruption of the “executive” attention network due to decreased functional connectivity, resulting in atrophy and processing impairments. In line with this notion, patients with FOG had significantly reduced scores on various tests related to executive function [66,67]. These findings also highlighted potential mechanisms that could serve as a target for novel treatments.

Cortical/subcortical pathway degeneration is different for each subtype of PD patients, which is known to be due to differences in their motor behavior. However, the effects of PD subtypes on cortical activity during walking are not yet well understood. In a study examining PD motor subtypes for cortical activity during walking using functional near-infrared spectroscopy (fNIRS), PIGD patients showed higher prefrontal cortical activity than TD patients [60]. Prefrontal cortical activity was higher in the PD group, even with the change in cortical activity during walking on a treadmill between the healthy control and PD groups [69]. These results indicate that PD patients need to recruit additional cognitive resources from the prefrontal cortex for gait. Subtyping of PD based on cortical activity is in the nascent stage, and only a few studies have attempted subtyping using fNIRS.

### 2.3. Molecular Biomarkers

#### 2.3.1. Genetic Markers

Several genetic factors, such as *SNCA*, *LRRK2*, *PRKN*, *PINK1*, and *GBA*, have been associated with PD [15], and new genetic loci that confer the risk for PD are still being discovered [70] (Table 3). Each mutation may result in a different pathogenic pathway and be vulnerable to specific molecular targets (similar to oncogenic mutations in carcinogenesis that have led to the development of matched anti-cancer drugs), but the evidence that the clinical symptoms and treatment outcomes of PD differ among molecular subtypes is still debatable [5,71,72]. A multi-modal clustering study combining clinical and genetic information from the Parkinson’s Progression Markers Initiative (PPMI) dataset demonstrated that the genetic risk score, defined by 30 PD-specific mutations, was of much less significance than clinical features [4]. Additionally, 90–95% of patients with PD are idiopathic, without known genetic etiology [73]. The precision medicine for PD subtypes based on genomic profiling requires more extensive studies on the role of genetics in the pathogenesis of PD. Clinical trials in genetically stratified patients with PD have only recently begun [74].

*LRRK2* is the most common PD-associated gene and is involved in multiple biological functions, such as mitochondrial signaling, vesicular trafficking, autophagy, and oxidative pathways [84]. Due to its versatility, we are still unaware of the function that has the most critical role in the pathobiology of PD [85]. The patients with PD harboring *LRRK2* mutation were clinically difficult to distinguish from idiopathic PD (iPD) patients [75]. Most phenotypes, with slight differences in tremor and non-motor features, overlapped between these two groups. Additionally, responses to dopaminergic treatment [77] and deep brain stimulation [86] were also similar in patients with *LRRK2* PD and iPD. However, according to a metabolomics study, *LRRK2* PD had a unique metabolomic profile compared to iPD [78]. Various studies focusing on the development of effective and safe drugs inhibiting *LRRK2* in *LRRK2* PD patients are ongoing [87,88]. Since *LRRK2* activity is also known to increase in iPD [89], it still remains to be evaluated whether the therapeutic effect of LRRK2-specific inhibitors will depend on the genetic subtype.

*PINK1* or *PRKN* mutations can induce dysregulation in mitochondria and are known to be associated with the risk of PD [90,91]. However, it is still unclear whether patients with PD harboring *PINK1* or *PRKN* mutations have a different neuropathology compared to others [92]. However, clinical studies examining the effect of co-enzyme Q10 or vitamin K2, which support electron transfer and ATP production in mitochondria, are underway in genetically stratified PD patients [72].

Glucocerebrosidase, encoded by another PD-causing gene, *GBA*, is another target of genotype-based PD therapy [93]. As one of the lysosomal hydrolases, glucocerebrosidase catalyzes the hydrolysis of glucosylceramide to ceramide and glucose. Considering that *GBA* mutation leads to a decrease in the activity of glucocerebrosidase, resulting in the accumulation of glucosylceramide, glucosylceramide synthase inhibitors are currently undergoing a clinical trial to treat patients with PD harboring *GBA* mutations [74]. Conversely, as glucocerebrosidase can reduce α-synuclein formation [94], upregulating the expression level of glucocerebrosidase is also being investigated for treating PD patients without *GBA* mutations [95].

In addition to causal mutations, single-nucleotide polymorphisms may result in endophenotypes of PD. Patients with the rs356182 *SNCA* single-nucleotide polymorphism were associated with the TD subtype and slow progression of motor symptoms [96]. PD with severe *GBA* variants showed faster disease progression, increased risk of dementia, and more rapid decline in cognitive function than PD with mild *GBA* variants [79,97]. PD with severe *GBA* variants was also associated with hallucinations [80].

Contrary to hopes for gene-specific therapies, the penetrance of monogenic forms of PD is incomplete [5,98]. For example, the penetrance of *LRRK2* mutations in PD is incomplete [99], and the age of onset or clinical phenomenology is different in individuals with the same mutation of *LRRK2* [100]. Other genetic, biological, and environmental factors should be considered to adequately subcategorize PD patients.

#### 2.3.2. Biochemical Markers

Since CSF is found within the tissue that surrounds the brain and spinal cord, it could reflect the molecular changes in the brain. In contrast to the neuropathology of PD, i.e., the aggregation of α-synuclein, leading to Lewy bodies, the level of α-synuclein in CSF was lower in patients with PD than in HCs; moreover, its association with clinical features showed a discrepancy between studies [16]. Other components related to Lewy body formation, including Aβ42, total tau, and phosphorylated-tau (p-tau), were also identified to be different in the CSF of patients with PD when compared to that of HCs [16]. These CSF fluid biomarkers might have a role in the early detection of PD [72].

Molecular-driven subtyping may result in a stratification that is different from the clinical-driven subtyping [101]; however, most studies have only attempted to characterize molecules between clinical-driven subtypes. Although the levels of total tau in CSF were significantly different in PD subtypes classified by clinical features, there was not much difference in serum markers [4]. On comparing PD subtypes, there was a decrease in apolipoprotein A1 and an increase in CRP levels of CSF in the severe motor subtype of PD [48]. The CSF of the diffuse-malignant PD subtype, associated with fast cognitive decline, showed an AD-like profile comprising low amyloid-β and amyloid-β/t-tau ratio [4]. The lower levels of amyloid-β and p-tau were associated with the subtype demonstrating postural instability and gait disturbances [102], while the lower level of α-synuclein in CSF was associated with the non-tremor-dominant subtype of PD [83].

Immune system dysfunction has been recognized as another hallmark of PD pathology [103]. The interplay of the central and peripheral immune systems can synergistically drive the initiation and progression of PD [104]. Immune-related biomarkers in the blood were investigated for subtyping PD since the concentrations of cytokines [105] and inflammatory molecules [106] correlated with the motor severity of the disease. Using a serum immune marker profile, patients with PD were classified based on the scores of the proinflammatory or anti-inflammatory component [82]. When classified according to proinflammatory components, the patients with a higher score were associated with lower cognitive function, measured by the Mini Mental State Examination. Meanwhile, when classified according to anti-inflammatory components, the patients with a higher score were associated with slower motor progression. These results were comparable with the aforementioned study [48] reporting that the PD subtype with severe motor and non-motor symptoms was associated with a proinflammatory biomarker profile with reduced apolipoprotein A1 and increased CRP.

Several biochemical markers are associated with neuroimaging features. From the network analysis of resting-state fMRI, the level of α-synuclein in CSF was correlated with decreased motor-related functional connectivity [107], and network disruption was correlated with both α-synuclein and Aβ42 [108]. The levels of α-synuclein and total tau in CSF were associated with microstructural changes on regional diffusion tensor imaging [109]. The structural connectivity analysis of patients with PD suggested that some network metrics, such as global efficiency and clustering coefficient, were associated with CSF levels of α-synuclein, Aβ42, and total tau [110]. Taken together, clinical, neurological, and biochemical features are interrelated, despite inconsistent results observed between certain studies.

#### 2.3.3. Transcriptomic Markers

With the advent of microarray and RNA-sequencing technologies, transcriptomic profiling has led to a better understanding as well as subtyping of diseases [111]. The genome-wide expression analysis for PD focuses on the identification of differentially expressed genes using the post-mortem brain [17]. In another prevalent neurodegenerative disease, AD, a recent study suggested five stable molecular subtypes using transcriptome data of 1543 cohort patients [112]. Each subtype demonstrated specific combinations of dysregulated pathways and driver genes. The subtype heterogeneity was also recapitulated in existing AD mouse models. However, further large-scale studies on PD subtypes based on transcriptome data are warranted.

As another approach, many studies have investigated the transcriptome signature in the peripheral blood of PD patients [17]. The blood–brain barrier is often compromised with age [113], and its impairment is widely observed in neurodegenerative diseases, including PD and AD [114,115]. Therefore, the blood of patients with PD may capture the pathobiology of the brain. Analysis of the transcriptome from the brain of living patients with PD consistently showed dysregulated PD biomarkers between the brain and blood compared to HCs; the downregulated genes were *ANKRD22*, *IL1R2*, *MARCH1*, and *OLFML2B*, while the upregulated genes were *BTNL9* and *STOX1* [116]. The differentially expressed genes (including *SYN1*, *GRIN1*, *GRIN2D*, and *DLGAP*)*,* enriched in the pathway of the neuronal system, were also higher in the blood of patients with PD [117]. Several studies on the post-mortem brain of PD also suggested *SYN1* as one of the PD biomarkers [118,119,120]. Among them, *SYN1* and *ANKRD22* were proposed as therapeutic response markers in blood samples of PD [121]. Additionally, the pathogenesis of PD, such as dysregulation of mitochondria, the ubiquitin–proteasome system, metabolism, and oxidative system, was captured in the transcriptome signature from the blood of patients with PD [122]. Such concordances imply that PD is not merely limited to neural tissue but is systemic in nature [17].

The relationship between transcriptomic alteration in the blood and clinical features has been investigated. Patients with rapid or slow PD progression have distinct transcriptomic signatures [123]. In PD patients, several biomarkers, such as *COPZ1*, *PTPN1*, and *MLST8*, were correlated with cognitive performance, measured by the Montreal cognitive assessment [124].

The expression of *SOD2*, *PKM2*, *ZNF134*, *ZNF160*, and *SLC14A1* was associated with the Unified Parkinson Disease Rating Scale score [124,125]. The *SNCA* level was related to cognitive decline in a longitudinal follow-up study of patients with PD [126]. Using next-generation small RNA sequencing, a study compared miRNA signatures between PD patients and HCs; the results suggested that the expression levels of multiple miRNAs in the CSF and blood were associated with the severity of Lewy body pathology [127]. These findings suggest that the molecular subtypes of PD can be matched to distinct clinical features. However, a study on the molecular subtypes of AD showed that classification by a genotype or biochemical biomarker is not enough to stratify the AD subtypes [112].

Molecular subtyping of PD is still in its initial stages, and, so far, only a few studies have been attempted [72,128]. The diagnostic potential of RNA biomarkers has been evaluated to distinguish PD from progressive supranuclear palsy, which is another parkinsonian syndrome classified as an atypical parkinsonian disorder [124]. Further studies are needed for PD subtyping using the transcriptomic signature.

## 3. Potential Subtyping of PD Using New Cluster-Associated Variables: Pilot Cluster Study

In our previous pilot assessor-blinded, randomized, controlled, parallel-group clinical trial using fNIRS, we found that acupuncture in PD patients led to an improvement in motor symptoms (gait disturbance) and rearrangement of the cerebral cortex, including the prefrontal cortex and supplementary motor area [129]. To investigate whether PD patients can be subtyped based on gait features, we preliminarily clustered participants (*n* = 26) according to gait parameters at the baseline before acupuncture treatment [129].

### 3.1. Clustering Method

K-means clustering using Euclidean distances was carried out in participants (*n* = 26) at the baseline before acupuncture treatment [130]. Based on data-driven analysis and the empirical opinions of the gait and fNIRS researchers, seven parameters related to gait and fNIRS results were evaluated. The selected gait features were velocity, cadence, stride time, stride length, single support, double support, and spatial symmetry. Because each feature had a different scale, we performed a Z-score transformation before cluster analysis to prevent specific features from having a dominant effect [4,43,131,132]. We also ruled out outliers with a magnitude of Z-score > 2.5 to improve the quality of the outcome [4,43,132,133]. Because there were no missing data, pre-processing for missing values was not required.

### 3.2. Clustering Results on Baseline Evaluation Using Gait Features in PD (Cross-Sectional Analysis)

Various clustering analyses were performed to estimate the optimal cluster number; we selected three clusters based on the silhouette coefficient of the cluster analysis. Silhouette coefficients for the clustering number (K) were 0.487 (K = 3), 0.483 (K = 4), and 0.333 (K = 5), respectively [133,134,135]. The clustering results are described in detail in Table 4. The three-dimensional scatter plot of the principal component analysis to distinguish each subtype is shown in Figure 2 [136,137].

Dimension reduction of the data was performed to reduce all features to three principal components, which were representative of all features. This technique is useful in visualizing the distribution of the clusters.

Patients in Cluster 1 showed more severe motor symptoms based on H&Y, UPDRSA, and UPDRSM than patients in other clusters. The velocity and cadence in the patients of Cluster 1 were the fastest. In contrast, patients in Cluster 0 showed the lowest velocity and cadence. Although the severity of motor symptoms in PD was classified for each cluster, it was difficult to analyze due to the limitation of the small sample size.

Because the proposed clustering was for a pilot study, we used a very small data set (*n* = 26) for the subtype clustering analysis. Although the expected reliability of the results was not high due to the small sample size, we verified the feasibility of the clustering method using statistical techniques (*p*-values). In the future, we plan to include a larger sample size to collect gait and fNIRS data to improve the quality of our clustering outcome and prove the applicability of the proposed method in a larger cohort.

### 3.3. Cluster Changes after 8 Weeks (Longitudinal Analysis)

After a follow-up period of 8 weeks, cluster changes within individual participants with PD were analyzed using the machine learning process. As illustrated earlier (Section 3.2), the cluster analysis revealed three clusters (Clusters 0, 1, and 2) on the baseline cross-sectional assessment. We built a machine learning model (random forest) that could predict the cluster changes within individual participants at 8 weeks post-clustering; the model was based on the training data from the baseline clustering analysis. Figure 3 represents the machine learning process for cluster change prediction.

Four patients out of the total (*n* = 26) shifted from the baseline cluster group to the other group after 8 weeks of follow-up (Table 5). However, the reliability of our clustering and classification method employing a random forest algorithm may not be so high due to the insufficient sample size. Additionally, there was an acupuncture intervention in some of the participants during the 8-week post-baseline analysis. Despite the limitations, our results would provide a basis for further large-scale clinical trials for clustering and subtyping PD using gait features.

## 4. Conclusions

PD is considered a social problem as it deteriorates the patients’ quality of life and increases their financial burden [1]. Considering the clinical heterogeneity of PD, here, we summarized its subtypes using the clinical symptoms, neuroimaging, and molecular markers from previous studies for improving treatment efficiency through personalized medicine. The clinical subtyping of PD is mainly based on motor symptoms: TD and PIGD subtypes. The PIGD subtype shows severe clinical symptoms such as changes in spatiotemporal parameters during gait [25,26,27,28,29,30], greater loss of functional connectivity in the cerebellum [36], and lack of an endogenous defense system to prevent oxidative stress from damaging and destroying dopaminergic cells in the substantia nigra [38]. In subtype classification studies using both motor and non-motor features of PD, non-motor symptoms, including depression and anxiety, were more often observed in patients with cerebellar gray matter atrophy [45]. Patients with severe motor symptoms, prominent non-psychiatric disorders [43], and a diffuse/malignant subtype showed rapid disease progression [4,43], reduced survival [46], atrophy in PD-specific brain networks, and an AD-like CSF profile [4]. Taken together, the PD subtype with severe motor and non-motor symptoms tended to have a poor prognosis.

Research has been ongoing to find neuroimaging-based biomarkers for the early diagnosis and monitoring of the progression of PD (Table 2). Dopaminergic PET and SPECT techniques can track the development of motor and non-motor symptoms in PD. MRI and diffusion tensor imaging techniques are advantageous in the differential diagnosis and classification of PD because they can detect the structural and functional changes in the brain. Structural MRI confirms the difference between non-PIGD and PIGD subtypes, and functional MRI confirms the underlying pathophysiological mechanisms through brain network connectivity patterns. Identifying the pathological substrate responsible for motor symptoms through neuroimaging may play an important role in clinical trials, especially in those evaluating therapeutic efficacy. In addition, there is limited evidence on cortical pathway degeneration in PD subtypes during walking. However, fNIRS has apparent benefits in clinical brain imaging as it allows observation of cortical activity during gait.

Based on various pathobiological studies exploring genetic mutations, different treatment strategies have been developed depending on the genetic background of patients with PD. Clinical trials targeting the subgroups of patients harboring *LRRK2* or *GBA* mutations are underway. There is still a lack of evidence on whether PD subtypes based on genetic mutations differ from those based on clinical symptoms. Most biochemical studies have attempted to correlate concentration levels of serum or CSF markers between subtypes of PD. The levels of total tau, p-tau, apolipoprotein A1, CRP, and amyloid-β in CSF correlated well with subtypes categorized based on the clinical symptoms. Moreover, the levels of cytokines and inflammatory molecules in the blood revealed an association with clinical symptoms. Some biochemical molecules were also related to neuroimaging features, such as the functional connectivity of fMRI data. While the molecular subtyping of PD using transcriptomic signatures is still in its nascent stage, the peripheral blood from patients with PD showed a distinct transcriptome profile compared to HCs, and the expression levels of specific genes showed an association with clinical symptoms.

Additionally, we identified three subtypes via preliminary clustering by clinical phenotype based on gait parameters in 26 participants at the baseline before acupuncture treatment. Therefore, this review and the results of the preliminary clustering study identify PD subtypes and encourage precision medicine therapeutic strategies for various neurodegenerative disorders. Furthermore, this study may be helpful in developing personalized medicine for incurable disorders.

## Figures and Tables

**Figure 1 diagnostics-12-00112-f001:**
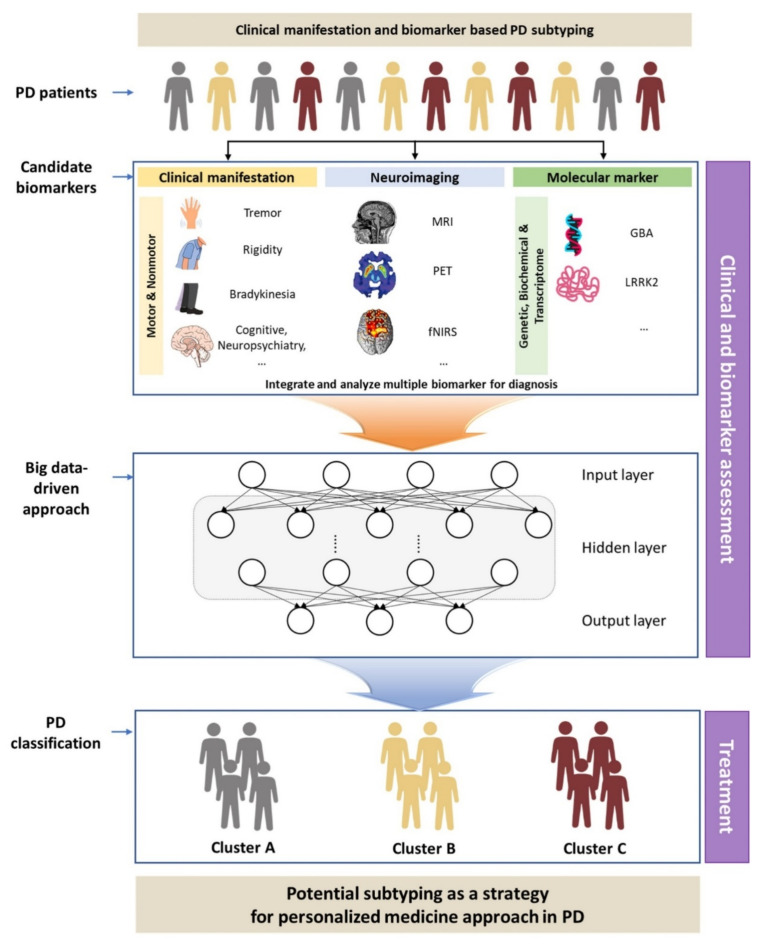
Schematic flow diagram of the classification for subtyping of Parkinson’s disease (PD). MRI, magnetic resonance imaging; PET, positron emission tomography; fNIRS, functional near-infrared spectroscopy.

**Figure 2 diagnostics-12-00112-f002:**
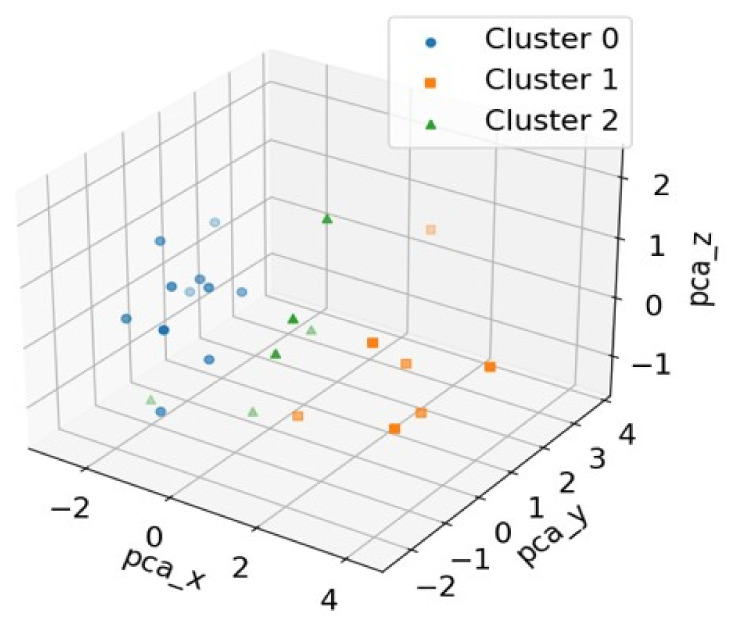
Three-dimensional scatter plot of the principal component analysis (PCA) to distinguish the three clusters.

**Figure 3 diagnostics-12-00112-f003:**
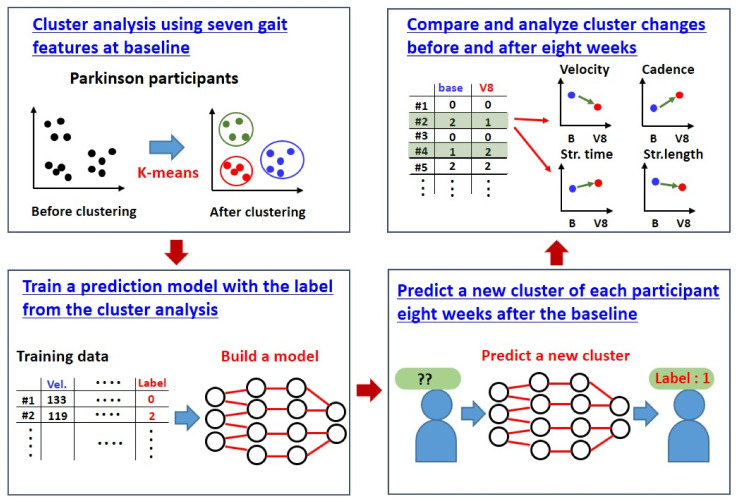
The cluster change prediction process with machine learning.

**Table 2 diagnostics-12-00112-t002:** Neuroimaging.

Criteria of Classifier	Classification of Subtypes	Classifier Profile	Findings	Reference
Structural imaging	PD with the PIGD PD with the non-PIGD	Multi-modal MRI scan in PIGD (resting-state fMRI, 3D T_1_-weighted MRI and DTI)	The classifier discriminated patients with the PIGD subtype with a diagnostic accuracy of 92.31%. Machine learning-based automatic classification	[55]
	PD with the PIGD PD with the non-PIGD	Diffusion tensor imaging in PIGD	Greater loss of white matter integrity in PIGD In particular, bilateral superior longitudinal fasciculus	[56]
	Neuroimaging cluster by cortical atrophy patterns pattern 1 (*n =* 33): cortical thinning in bilateral orbitofrontal, anterior cingulate, and lateral and medial anterior temporal gyri pattern 2 (*n* = 44): cortical thinning in bilateral occipital gyrus, cuneus, superior parietal gyrus, and left postcentral gyrus	MRI (T1-weighted images in a 3-tesla Siemens scanner)	There is evidence of cortical brain atrophy in the early stages of PD. Neuroimaging clustering analysis is able to detect subgroups based on cortical thinning.	[57]
	Neuroimaging cluster by cortical atrophy patterns parieto-temporal pattern of atrophy with worse cognitive performance (pattern 1) occipital and frontal cortical atrophy with younger disease onset (pattern 2) non-detectable cortical atrophy (pattern 3)	Neuropsychological assessment MRI	Decline in processing speed (as measured by the Stroop Word-Color test, the Symbol Digits Modalities test and the Trail Making Test Part B) and in semantic fluency in pattern 2, 3, and HC Greater compromise in activities of daily living and suffered higher attrition rate in pattern 1	[58]
	Mild-motor-predominant Intermediate-malignant Diffuse-malignant	DTI	MD of globus pallidus was associated with worsening of motor severity, cognition, and GCO. Baseline MD of nucleus accumbens, globus pallidus, and basal ganglia were linked to clinical subtypes	[59]
Functional imaging	TDPIGD	fNIRS, EEG, and gait parameters	PFC activation ↑ in PIGD than TD patients, regardless of the walking condition. Alpha and beta power in the FCz and CPz ↓ in both TD and PIGD. PIGD patients need to recruit additional cognitive resources from the PFC for walking.	[60]
	Young adults (YA) Older adults (OA) PD	fNIRS	PFC activity can be acceptably reliable and can differentiate young, older, and PD groups. PFC activation ↑ in PD than in young and older people during walking.	[61]
	PDNC	MRI (complex networks for accurate early diagnoses)	Connectivity of several brain regions is significantly related to PD. Provide a diagnostic index using complex network features with clinical scores	[62]

DTI, diffusion tensor imaging; MD, mean diffusivity; GCO, global composite outcome; fNIRS, functional near-infrared spectroscopy; PFC, prefrontal cortex; NC, normal controls.

**Table 3 diagnostics-12-00112-t003:** Molecular subtypes.

Criteria of Classifier	Classification of Subtype	Participants (Number)	Findings	Reference
Genetic	*LRRK2* PD, iPD	*LRRK2* PD (*n* = 25) iPD (*n* = 84)	*LRRK2* PD was associated with more tremor and better olfactory identification than iPD.	[75]
	*LRRK2* PD, iPD	*LRRK2* PD (*n* = 97) iPD (*n* = 391)	*LRRK2* PD was associated with lower extremity involvement at onset and PIGD score.	[76]
	*LRRK2* PD, iPD	*LRRK2* PD (*n* = 25) iPD (*n* = 84)	No differences between motor progression in *LRRK2* PD and iPD.	[77]
	*LRRK2* PD, iPD	*LRRK2* PD (*n* = 12) iPD (*n* = 41)	Metabolomic profiles distinguished patients with PD harboring *LRRK2 G2019S* mutation from patients with iPD	[78]
	*GBA* mutant PD, *GBA E326K* PD, Not *GBA* PD or *GBA* variants (mutant + *E326K*) PD, Not *GBA* PD	*GBA* mutant PD (*n* = 27), *GBA E326K* PD (*n* = 31), not *GBA* PD (*n* = 675)	*GBA* variants PD was associated with more rapid decline in UPDRS III score. *GBA* variants and *E326K* PD was associated with faster progression in PIGD scores. *GBA* variants and *E326K* PD was associated with progression to mild cognitive impairment or dementia.	[79]
	Mild, severe, risk, or complex *GBA* PD, Not *GBA* PD	Mild *GBA* PD (*n* = 32), severe *GBA* PD (*n* = 36), risk *GBA* PD (*n* = 21), or complex *GBA* PD (*n* = 16), not *GBA* PD (*n* = 27)	*GBA*-PD was associated with earlier and more frequent occurrence of several non-motor symptoms. Severe and complex *GBA*-PD was associated with the highest burden of symptoms and a higher risk of hallucinations and cognitive impairment. Complex *GBA*-PD was associated with the lowest β-glucocerebrosidase activity.	[80]
	*GBA* PD, Not *GBA* PD	*GBA* PD (*n* = 33), not *GBA* PD (*n* = 27)	*GBA* PD was associated with more rapid disease progression of motor impairment and cognitive decline, and reduced survival rates.	[81]
Biochemical	Higher proinflammatory score group, lower proinflammatory score group, or higher anti-inflammatory score group, lower anti-inflammatory score group	PD (*n* = 230) was dichotomized at the mean into high- and low-score groups	Higher proinflammatory and lower anti-inflammatory score groups were associated with more rapid motor progression. Higher proinflammatory score group was associated with lower MMSE.	[82]
	Quintiles for CSF biomarker levels, or TD and non-TD	PD (*n* = 660) was classified into quintiles (biomarker level 0–20, 20–40, 40–60, 60–80, and 80–100 percentile), or TD (*n* = 293) and non-TD (*n* = 118).	PD with the lowest amyloid-β level, the highest total tau/amyloid-β ratio, and the highest total tau/α-synuclein quintiles were associated with severe non-motor dysfunction.The CSF level of α-syn was significantly lower in non-TD.	[83]

PD, Parkinson’s disease; iPD, idiopathic PD; PIGD, postural instability and gait difficulty; UPDRS, unified Parkinson’s disease rating scale; MMSE, mini mental status examination; TD, tremor-dominant; CSF, cerebrospinal fluid.

**Table 4 diagnostics-12-00112-t004:** Mean and standard deviation of the seven features with three clusters.

Feature	Cluster 0 (*n* = 11)	Cluster 1 (*n* = 7)	Cluster 2 (*n* = 6)	*p*-Value ^†^	Post-Hoc ^‡^
Men, *n* (%)	6 (66.67%)	5 (62.50%)	4 (57.14%)	0.9999 ^††^	-
Women, *n* (%)	3 (33.33%)	3 (37.50%)	3 (42.86%)	-	-
Age (years)	64.33 ± 6.44	60.88 ± 7.30	63.57 ± 11.80	0.6948	-
Age at onset (years)	57.22 ± 7.41	50.63 ± 9.77	58.14 ± 12.89	0.2882	-
Disease duration (years)	7.11 ± 3.48	10.25 ± 5.73	5.43 ± 2.23	0.0912	-
Hoehn and Yahr scale score	1.67 ± 0.50	2.25 ± 0.71	1.57 ± 0.53	0.0669	-
UPDRSA	1.33 ± 0.50	2.00 ± 1.07	1.86 ± 1.35	0.3650	-
UPDRSM	4.11 ± 2.37	6.13 ± 2.36	4.00 ± 1.29	0.1023	-
Velocity	105.35 ± 5.85	143.25 ± 8.76	116.63 ± 6.40	<0.0001 *	A < C< B
Cadence	112.00 ± 4.08	127.07 ± 7.50	129.06 ± 3.52	<0.0001 *	A < B,C
Stride time (s)	1.07 ± 0.04	0.95 ± 0.06	0.93 ± 0.02	<0.0001 *	C,B < A
Stride length (cm)	113.15 ± 6.08	135.68 ± 7.03	108.52 ± 5.10	<0.0001 *	C,A < B
Single support (s)	0.41 ± 0.02	0.37 ± 0.03	0.36 ± 0.01	0.0009 *	C,B < A
Double support (s)	0.27 ± 0.01	0.20 ± 0.03	0.21 ± 0.03	<0.0001 *	B,C < A
Spatial symmetry	36.69 ± 0.67	39.59 ± 1.07	38.66 ± 1.41	0.0031 *	-

Values are presented as means ± standard deviation. ^†^ Significant difference between the intervention and control group in one-way analysis of variance (ANOVA). ^‡^ Post-hoc test by Bonferroni procedure after one-way ANOVA. ^†^^†^
*p*-value were analyzed by Exact-test. * *p* < 0.05. UPDRSA, a subsection of the Unified Parkinson’s Disease Rating Scale (UPDRS) score that includes the “walking and balance” and “freezing” parts of the UPDRS II assessment for activities of daily living; UPDRSM, a subsection of the UPDRS score that includes the “gait”, “postural stability”, “posture”, and “body bradykinesia” parts of the UPDRS III motor assessment; A, Cluster 0; B, Cluster 1; C, Cluster 2.

**Table 5 diagnostics-12-00112-t005:** Gait parameter changes during 8 weeks for four participants with changed clusters.

#4	Velocity	Cadence	Stride Time (s)	Stride Length (m)	Single Support	Double Support	Spatial Symmetry	Cluster
Base	133.20	121.90	0.99	131.25	0.37	0.25	37.87	1
V8	112.10	105.93	1.13	127.30	0.42	0.29	37.35	0
#10	Velocity	Cadence	Stride time (s)	Stride length (m)	Single support	Double support	Spatial symmetry	Cluster
Base	146.50	134.57	0.89	130.54	0.35	0.19	39.00	1
V8	128.76	133.13	0.90	116.49	0.34	0.22	37.85	2
#13	Velocity	Cadence	Stride time (s)	Stride length (m)	Single support	Double support	Spatial symmetry	Cluster
Base	101.27	112.83	1.07	107.86	0.40	0.27	37.83	0
V8	132.37	129.38	0.98	129.38	0.39	0.21	39.45	1
#15	Velocity	Cadence	Stride time (s)	Stride length (m)	Single support	Double support	Spatial symmetry	Cluster
Base	111.57	116.17	1.03	115.39	0.38	0.29	36.33	0
V8	137.43	128.73	0.93	127.66	0.35	0.23	37.73	1

## Data Availability

The source document data used to support the findings of the present study have been deposited in the Clinical Trial Center of Daejeon Korean Medicine Hospital, Daejeon University repository (IRB number DJDSKH-17-BM-20).

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
