# Peer review of "Parkinson’s Disease Subtyping Using Clinical Features and Biomarkers: Literature Review and Preliminary Study of Subtype Clustering"

_diagnostics, 2022, doi:10.3390/diagnostics12010112_

Round 1

Reviewer 1 Report

Personalized medicine is considered as a promising approach to improve treatment efficiency in various disease entities ranging from cancer to neurodegenerative ailments. Although Parkinson’s disease (PD) is a common and well-studied neurodegenerative disorder, classifying disease subtypes according to clinical manifestations and biomarkers remains challenging due to the broad variety of symptoms and their heterogeneous presentation in patients. Thus, more studies attempting to define disease classes using various parameters are urgently needed.

In their manuscript entitled “Parkinson’s disease subtyping using clinical features and biomarkers: Literaure review and preliminary study of subtype clustering”, Lee and colleagues provide an extensive overview of previous studies on PD classifications. After introducing general features of PD, they first describe existing classifications according to clinical features using motor symptoms, non-motor symptoms or a combination of both. Then, they summarize studies using neuroimaging as classifier, followed by an overview on existing classifications defined by molecular biomarkers including genetic, biochemical, and transcriptomic markers.  Lastly, the authors report their results on a pilot cluster study using gait parameters and fNIRS data, were they identified three PD subtypes.

Overall, this review provides a comprehensive overview on the existing literature about PD classifications. The article is well-structured and benefits from several tables and well-conceived figures which help the reader to organize the different studies discussed in the text and to visualize the processes of subtyping PD patients. In addition, the authors present an alternative classification system using new cluster-associated variables, which might facilitate the development of further personalized medicine approaches in the future. However, there are also some points worth to be addressed to improve the overall quality of the manuscript. These are listed in detail in the following:

1) The pilot cluster study performed by Lee et al. provides an interesting novel approach using gait features to classify PD subgroups. However, as the authors point out, the case number is very low and thus it is important to bear in mind that the applicability of this method needs to be proven in a larger cohort.

2) The identified groups (clusters 0, 1 and 2) are poorly characterized in the text and therefore remain very abstract for the reader. Thus, it would be beneficial to summarize the characteristic features of each cluster in more detail, for example: Patients in Cluster 0 show reduced velocity and cadence, but increased stride time compared to Cluster 1 and 2 etc. Furthermore, it would be interesting to see how the findings of the clustering analysis compare to results from a conventional approach using common classifiers, such as motor symptoms; for example: Patients in Cluster 0 show more severe motor symptoms than patients in Cluster 1 etc.

3) Moreover, it would be important to know more details about the patient characteristics and therefore heterogeneity of participants such as inclusion criteria, age, sex, age of disease onset, H&Y, UPDRS.

4) Please check if all abbreviations are introduced adequately. Some abbreviations are neither explained sufficiently in the text nor in the tables and table legends, for example IMU sensor, ROM, MMSE, ReHo, SFT Animal, HVLT-R (all Table 1).

5) Table 3 lists studies identifying subtypes based on molecular biomarkers. While many studies using genetic modifiers are included, only one study using biochemical parameters is mentioned in the table although more studies are discussed in the text. Thus, I would recommend adding some more biochemical marker studies to the table.

6) In general, the language style used in the manuscript is appropriate, however, there are some grammatical errors and some sentences are hard to follow. In these instances, minor language editing would be beneficial. Please find some examples here:

  • Introduction, 4th paragraph, 1st word: “Various” instead of “arious”
  • Introduction, last paragraph: “…and the number of failed attempts to establish a simplistic single-target approach to the drug for the therapy of PD…”. I would recommend revising this sentence as it is hard to follow.
  • Introduction, last paragraph: remove the article before “PD occurrence”and “clinical symptoms”
  • Introduction, last paragraph, last sentence: remove “.” after “Figure”
  • Section 2: replace “reviewed” by “review”
  • Head line 2.1: replace “feature” by “features”
  • Section 2.1: revise the following sentence, divide into 2 sentences if possible: “Previous studies classifying PD subtypes…, was classified using motor symptoms as the dominant criteria.”
  • Section 2.1.1, 2nd line: remove “.” after “Table”
  • Section 2.1.1, 2nd paragraph: replace “the” before “unified Parkinson’s disease rating scale” with “a higher”
  • Section 2.1.1, 2nd paragraph, last sentence: insert “other” after “various”
  • Section 2.1.1, 3rd paragraph: replace “higher” cerebellum disruptions with “more severe”
  • Section 2.1.3, 4th paragraph: replace “predominantly” motor manifestations with “predominant”
  • Section 2.1.3, 4th paragraph: shift “,” after “at post mortem”
  • Section 2.2, 4th paragraph: remove the article before “early PD patients”
  • Section 2.2, 7th paragraph, last sentence: replace with “…could serve as target for novel treatments”.
  • Section 2.3.1, first sentence: remove the article before “new genetic loci”
  • Section 2.3.3, last paragraph: replace “by far” with “so far”
  • Section 3, first head line: incorrect section number (2.4)
  • Section 3, 3rd paragraph: remove “.” after “Figure”
  • Section 3, 3rd paragraph, last sentence: insert “the” before “future”
  • Section 3, 4th paragraph, 2nd sentence: there is no section 3.2 in the manuscript
  • Section 3, 4th paragraph: remove “.” after “Figure”

7) Please check consistent use of medical terms, for example section 2.3.2, last paragraph: a-synuclein vs. a-synuclein.

Reviewer 2 Report

13th December, 2021

Review of the Manuscript ID: diagnostics-1483038, by S. H. Lee et al., entitled: “Parkinson's Disease Subtyping using Clinical features and Biomarkers: Literature review and Preliminary study of Subtype Clustering” that is intended to be published as the Review in Diagnostics

(separate Microsoft Word file as Reviewer Attachment for Manuscript ID diagnostics-1483038 Diagnostics 13th December 2021 that includes Comments to the Authors is also uploaded)

Taking into account research highlight, contribution of the Authors to the progress in the research area, precise manner of data presentation, very well writing in English, abundance of Results. Tables and Figures (thorough tabular documentation and diligent graphic visualization), the quality of this paper deserves praise and merits my support. The Authors have received the high scores from me for the originality, importance of the work and the scientific value of their paper. In my opinion, the current paper provides insightful interpretation of topical and coming trends in accurately identifying biomarkers and appropriately distinguishing the multi-faceted etiopathogenesis and neurodegenerative symptoms of Parkinson’s disease (PD). This could give rise to not only more comprehensive understanding of the heterogeneity noticed for clinical manifestations in PD, but also detailed classification of PD subtypes on the basis of their clinical features, neuroimaging, and specific biomarkers. Finally, this could bring about the development and optimization of the novel approaches for personalized medicine aimed to properly diagnose the multiple PD subtypes and thereby elaborate efficient therapeutic strategies targeted at the potent treatments of a variety of PD subtypes. For all these reasons, I strongly recommend the Editorial Board to allow for publication of this tremendously valuable paper in Diagnostics, after the minor revision of the manuscript will have been completed by the Authors and provided that the Authors are ready to consider all the Reviewer comments shown below:

1) There is a lack of the separate Abbreviations section in the paper. Therefore, this section should have been added at the end of the manuscript to thoroughly elucidate and expand a broad spectrum of the in-text abbreviations, which have been used by the Authors in all the subsections of their paper.

2) The References section has to be prepared in the format compatible with the requirements of Diagnostics.

General Comment of the Reviewer:

Before the manuscript will have been accepted for publication in Diagnostics, it requires the minor revision (according to all the recommendations and suggestions indicated above by the Reviewer).
